# Test–Retest Reliability of an Isometric and Isometric/Vibratory Muscular Strength Protocol with Functional Electro-Mechanical Dynamometry

**DOI:** 10.3390/sports12070175

**Published:** 2024-06-26

**Authors:** Oscar Andrades-Ramírez, David Ulloa-Díaz, Bryan Alfaro Castillo, Patricio Arroyo-Jofré, Antonio Castillo-Paredes, Luis Chirosa-Ríos

**Affiliations:** 1Facultad de Salud y Ciencias Sociales, Escuela de Ciencias de la Actividad Física, Universidad de las Américas, Concepción 4030000, Chile; 2Department of Sports Sciences and Physical Conditioning, Universidad Católica de la Santísima Concepción, Concepción 4030000, Chile; dulloa@ucsc.cl; 3Facultad de Educación, Universidad de Atacama, Copiapó 1531772, Chile; bryan.alfaro@uda.cl; 4Facultad de Educación, Universidad San Sebastián, Santiago 8420524, Chile; patricio.arroyo@uss.cl; 5Grupo AFySE, Investigación en Actividad Física y Salud Escolar, Escuela de Pedagogía en Educación Física, Facultad de Educación, Universidad de las Américas, Santiago 8370040, Chile; acastillop85@gmail.com; 6Department of Physical Education and Sport, Faculty of Sport Sciences, University of Granada, 18011 Granada, Spain; lchirosa@ugr.es

**Keywords:** muscular strength, reliability, dynamometer, isokinetic

## Abstract

The purpose of the study was to analyze the test–retest reliability of an isometric and isometric/vibratory muscular strength protocol in the bilateral seated bench press (BSBP), bilateral seated rowing (BSR), unilateral seated right knee extension (USKER), and left knee extension (USKEL) tests controlled using functional electromechanical dynamometry (FEMD) in healthy young adults. A repeated measures design was used to determine the reliability of a muscular strength protocol in isometric and isometric vibration modes with FEMD. No significant differences were found in test–retest analysis (*p* > 0.05; ES < 0.20); and high reliability (CV = 4.65–5.02%; ICC = 0.99–0.98) was found for BSBP measures, and acceptable reliability (CV = 3.71–9.61%; ICC = 0.98–0.95) was found for BSR, USKER, and USKEL. Furthermore, the coefficients between the two measures were strong (r = 0.963–0.839) and highly significant (*p* = 0.001) for maximal strength in the isometric and maximal isometric/vibratory assessment of muscle strength in all muscle strength tests. This study demonstrates that isometric and maximal isometric/vibratory strength in the BSBP, BSR, USKER, and USKEL tests can be measured with high reliability and reproducibility using the FEMD.

## 1. Introduction

Sports medicine, rehabilitation, and sports performance continue to be very interested in the systematization of reliable processes and protocols for the assessment of the many manifestations of muscular strength [1,2,3,4,5,6,7]. This allows for (i) better analysis and precision in the evaluation of training systems and changes in the level of physical performance that induce muscular strength training, and (ii) favoring and optimizing the function of coaches and trainers when providing feedback during the training process and competition [8].

Muscle contractions, whether dynamic or isometric, can be used to measure the peak force that can be produced; this capacity is continually observed in different age groups [9]. Equipment reliability testing is necessary due to the increasing popularity of isometric testing as a means of determining maximum strength and the ability to apply peak force in the shortest possible time [10]. It is crucial to employ evaluations that can distinguish between different types of analytical advancements in force creation when measuring force [11]. When it comes to training and assessing the kinetic variables of power, speed, and strength of various human movements [12], as well as the various forms of muscular contraction and strength manifestations in free, mono-, or multi-joint movement of the entire body [13], functional electromechanical dynamometry (FEMD) is a perfect tool. When compared to the gold standard, or the isokinetic device, the FEMD is especially easy to use and inexpensive. It can operate in two modes: dynamic (tonic, kinetic, elastic, inertial, conical) or static (isometric, vibratory), making it easier to evaluate and train using a constant and variable resistance/velocity [12,14].

In a recent study by Morenas-Aguilar et al. [15] that analyzed the relative and absolute reliability of a functional test with FEMD in three exercises related to strength in sprinting and throwing in handball in fourteen male handball players from the same Spanish first-division team, the results showed that all exercises reported acceptable to high reliability (ICC > 0.83–0.92) for average and maximum strength, in addition, they have a coefficient of variation (CV) < 11.97%. The results showed that all exercises for mid and peak strength had acceptable to high reliability (ICC > 0.83–0.92) with CV < 11.97% percent. According to Baena-Raya et al. [16], the FEMD for an isometric mid-thigh pull-up exercise shows high reliability in maximal strength (CV = 2.22–2.51%; ICC = 0.94–0.95) for twenty-seven male collegiate athletes with more than 6 months of structured resistance training experience regularly performing weightlifting movements. In the Jerez-Mayorga et al. study [14] using three incremental loads controlled by an FEMD in twenty-eight healthy elderly women, high reliability (ICC = 0.95–0.98) and stable repeatability were established for the protocols used for the strength and velocity of movement of the concentric phase of five Sit-to-Stand measures (CV < 10%). There is, however, no study that has examined the relative and absolute reliability for knee push, pull, or extension exercises in an isometric or isometric/vibratory seated mode. Instead, the relative and absolute reliability of studies with FEMD have only been demonstrated thus far in exercise protocols for intercession comparison inter-session (test–retest) in tonic, linear isokinetic, and isometric mode.

Therefore, the aim of this study was to analyze the absolute and relative test–retest reliability of the FEMD for an isometric and isometric/vibratory peak muscle strength protocol in the bilateral seated bench press, bilateral seated rowing, and seated unilateral right and left knee extension in healthy young men.

## 2. Materials and Methods

### 2.1. Study Design

To meet the study’s objective, a repeated measures design was used, and bilateral seated bench press (BSBP), bilateral seated rowing (BSR), and unilateral open kinetic chain seated knee extension tests were performed for the right (USKER) and left (USKEL) leg in isometric and isometric/vibratory mode with FEMD to analyze peak muscle strength. After two FEMD familiarization sessions, participants came to the laboratory for data collection on two days separated by at least 48 h. Participants assessed their peak muscular strength for three repetitions lasting five seconds (s) on each of these days. Every evaluation was conducted at the undergraduate physical activity science laboratory at the Universidad de las Américas Concepción, Chile, at the same time (h) of the day (±1 h) and with identical ambient conditions (≈23 °C and ≈60% humidity).

### 2.2. Subjects

Twelve (*n* = 12) physically active college students (age 22.25 ± 1.03 years, weight 71.40 ± 8.16 kg, height 1.71 ± 0.18 m) without any experience with isokinetic devices or dynamometers voluntarily participated in the study. The participants (a) did not present any musculoskeletal pathology and (b) did not practice specific upper body strength training.

Before providing their written agreement, each research participant was made aware of the purpose, nature, and hazards of the experimental technique. The Universidad Católica de la Santísima Concepción No. 47/2023 (approved 22 November 2023) ethics and research committee authorized the study protocol, which was carried out in accordance with the Declaration of Helsinki [17].

### 2.3. Materials

The isometric and isometric/vibratory force was evaluated with an FEMD (Dynasystem, Model Research, Granada, Spain) with a precision of 3 mm for the displacement, 100 g for the detected load, and a sampling frequency of 1000 Hz and a range of velocities between 0.05 m·s^−1^ and 2.80 m·s^−1^. A wide variety of movements can be assessed in different anatomical planes, and the device can deliver a wide variety of stimuli (isokinetic, isotonic, elastic, isometric, inertial, eccentric, and vibratory) for the assessment of muscle strength in its different manifestations. Its core control precisely regulates both force and angular velocity using a 2000 W electric motor. The user applies forces to a rope wound on a roller, controlling and measuring both force and linear velocity. A load cell detects the tension applied to the rope, and the resulting signal is passed to an analog-to-digital converter with 12-bit resolution. The displacement and speed data are collected with a 2.500 ppr encoder attached to the roller. The data from the different sensors are obtained at a frequency of 1 kHz.

### 2.4. Protocol for Familiarization

Prior to the intervention, all participants underwent two sessions of familiarization with the use of the FEMD and the four isometric and isometric/vibratory strength assessment exercises to be evaluated for the duration of the 45 min session. The activity began with a standardized warm-up in two parts. The first considered activation during 5 min of static cycling at low intensity (50–65% of heart rate reserve), and the second part of the warm-up consisted of 3 sets of 10 repetitions with a load of 10% of their body weight for each muscle strength assessment.

### 2.5. Assessment of Muscular Strength

After the standardized warm-up phase, subjects performed isometric and isometric/vibratory strength testing with a vibratory rate of 20 (Hz/mm) and 40 (Hz/mm) peak in 3 sets of 5 s with a 3 min pause between measurements until reaching the 1RM using the FEMD in 4 exercises, as shown in Figure 1: (a) bilateral seated bench press with shoulder abduction at 90° and elbow flexion at 90°, (b) bilateral seated rowing with shoulder abduction at 90° and elbow flexion at 90°, and (c) unilateral seated knee extension in open kinetic chain with 90° knee flexion. 

### 2.6. Statistical Analysis

Descriptive statistics were used to calculate the means and standard deviations (SD) for the maximal muscular strength assessment tests. The normal distribution of the data was analyzed using the Shapiro–Wilk test. The data were obtained in the normality test (*p* > 0.05). Paired-samples *t*-test and standardized mean differences (effect size for repeated samples) were used to compare peak strength between repetitions. The criteria for interpreting the magnitude of the effect size (ES) were as follows: null (<0.20), small (0.2–0.59), moderate (0.60–1.19), large (1.20–2.00), and very large (>2.00) [18]. Absolute reliability was assessed using the standard error of measurement (SEM) and coefficient of variation (CV), and relative reliability was assessed using the intraclass correlation coefficient (ICC) model. The CV was obtained by dividing the standard deviation by the mean in the absolute value and multiplying the result by 100 to express it as a percentage. The following criteria were used to determine acceptable (CV ≤ 10%) and high (CV ≤ 5%) reliability [19]. The relative reliability (ICC) was classified as follows: values close to 0.1 were low, 0.3 was moderate, 0.5 was high, 0.7 was very high, and values close to 0.9 had extremely high reliability [20]. Bland–Altman plots were constructed to explore the agreement of FEMD with respect to muscular strength assessments and to quantify systematic bias and 95% limits of agreement between series [21]. The heteroscedasticity of the errors in the Bland–Altman plots and their 95% limits of agreement (Limits of agreement [LoA] = bias ± 1.96 SD) was defined as a coefficient of determinacy (R^2^) < 0.1 [22]. Finally, Pearson’s correlation coefficient (Pearson’s r) was used to quantify the correlation of all outcome variables between both evaluation sessions. The criteria for interpreting the magnitude of r were null (0.00–0.09), small (0.10–0.29), moderate (0.30–0.49), large (0.50–0.69), very large (0.70–0.89), almost perfect (0.90–0.99), and perfect (1.00) [23]. For all statistical calculations, a 95% confidence interval was used in the analysis. Statistical significance was accepted at *p* < 0.05. Relative and absolute reliability assessments were performed using a customized spreadsheet [18], while other statistical analyses were performed using JASP software (version 0.16.4).

## 3. Results

No significant differences were found in the test–retest analysis in isometric and peak isometric/vibratory strength assessments (*p* > 0.05; ES < 0.20). Reliability is reported as high (CV = 4.65%; ICC = 0.99) for the BSBP measures and acceptable (CV = 9.61–6.37%; ICC = 0.98–0.92) for the BSR, USKER, and USKEL measures, and it is observed that the SEM varied between 3.61 and 5.02 for peak strength in the isometric strength assessment, as shown in Table 1.

Reliability is reported as high (CV = 4.95%; ICC = 0.96–0.95) for the BSBP measurements and acceptable (CV = 8.38%; ICC = 0.97–0.95) for the BSR, USKER, and USKEL measurements in the isometric/vibratory force assessment of vibratory index 20 (Hz/mm), and it was found that the SEM varied between 3.59 and 5.01 for the peak force, as shown in Table 2.

For the evaluation of the isometric vibratory force of vibratory index 40 (Hz/mm), the BSBP, USKER, and USKEL measurements reported high reliability (CV= 5.02–3.71%; ICC= 0.98), and it was acceptable for BSR (CV = 7.17%; ICC = 0.96). It is observed that the SEM varied between 3.46 and 5.35 for the peak force, as shown in Table 3.

The Bland–Altman plots reveal a low systematic bias (−3.128–−0.089) for isometric assessment, isometric/vibratory vibratory rate 20 (Hz/mm), and isometric/vibratory rate 40 (Hz/mm) for peak force; in addition, a coefficient of determination R^2^= 0.953–0.733 was found, as shown in Figure 2.

In the correlation analysis of the magnitude evaluation r was categorized almost perfect (0.963–0.942) for the BSBP and USKER measurements, and for the BSR and USKEL measurements, it was categorized very large (0.856–0.869) for the isometric evaluation. The correlations in the vibrational isometric force 20 vibrational index for the BSBP, USKER, and USKEL measurements were almost perfect (0.930–0.976), and they were very large for BSR (0.887). For the isometric vibratory force 40 vibratory index, the correlations were almost perfect (0.951–0.954) for BSBP, USKER, and USKEL and very large for BSR (0.839), as seen in Figure 3.

## 4. Discussion

The purpose of this study was to analyze the absolute and relative test–retest reliability of the FEMD for an isometric and isometric/vibratory peak muscle strength protocol in the bilateral seated bench press, bilateral seated rowing, and seated unilateral right and left knee extension in healthy young men. The main results of this study showed an acceptable absolute reliability for all strength assessments (CV < 10%), an SEM = 3.46–5.35, and an extremely high relative high reliability (ICC = 0.92–0.99), with no significant differences reported between both assessment sessions with a null effect size. Furthermore, the Bland–Altman plots reveal a low systematic bias (−3.128–−0.089), with the coefficient of determination R^2^ = 0.953–0.733 possessing values close to a perfect fit. The association of the measure was categorized as very large to almost perfect (r = 0.839–0.976) in all isometric and isometric/vibratory peak muscle strength assessments.

Multi-articular isokinetic devices can be reliably used to assess and train certain movement and muscle activation patterns, as demonstrated by FEMD [23]. Morenas-Aguilar et al. [15] reported reliability measures similar to our study, showing the absolute and relative reliability of three specific strength tests in handball players with FEMD (stan-ding lift, unilateral pullover, and step forward). They observed an acceptable absolute reliability for the peak strength of the standing lift, unilateral pullover, and step forward (CV = 3.90–11.57%) and a relative reliability between very high and extremely high (ICC = 0.83–0.97). Rodriguez-Perea et al. [24] showed reliability measures close to but lower than our study, in which they evaluated the peak isokinetic strength of the trunk flexors with FEMD, obtaining an acceptable relative reliability (ICC = 0.71–0.83) in the measurements of the trunk flexors for concentric contraction, as well as an acceptable relative reliability for eccentric contraction (ICC = 0.74–0.87). In addition, the absolute reliability obtained better results in eccentric contraction (CV = 5.70–7.76%) than in concentric contraction (CV = 7.04–14.00%). In addition, in that study, an isometric evaluation of peak strength was used, reporting a very high relative reliability (ICC = 0.71–0.85) and an acceptable absolute reliability (CV = 6.82–10.83%). In another study, Martinez-Garcia et al. [25] evaluated shoulder joint strength with FEMD, and a null effect size (−0.19–0.10), acceptable absolute reliability (CV = 8.27%), and very high relative reliability (ICC = 0.85) were observed for the concentric phase and also for the eccentric phase (CV = 7.28%; ICC = 0.81) at a speed of 0.3 m·s^−1^. In the concentric phase, an acceptable absolute reliability (CV = 6.31%) and an extremely high relative reliability (ICC = 0.93) were obtained. In the eccentric phase, an absolute reliability (CV = 6.87%) and a very high relative reliability (ICC = 0.87) were reported for the internal rotator muscles of the shoulder joint. For the external rotators of the shoulder musculature, an acceptable absolute reliability (CV = 6.91%) and an extremely high relative reliability (ICC = 0.98) were reported for the concentric phase, and, for the eccentric phase, an acceptable absolute reliability (CV = 6.39%) and an extremely high relative reliability (ICC= 0.90) at a speed of 0 were reported. In the concentric phase, the values obtained for the absolute reliability were considered acceptable (CV = 6.26%) and high for the relative reliability (ICC = 0.89). For the eccentric phase, a high absolute reliability (CV = 5.12%) and an extremely high relative reliability (ICC = 0.92) at a speed of 0.6 m·s^−1^ were reported. In the study by Baena-Raya et al. [16], they analyzed the reliability of mid-thigh traction kinetic variables with FEMD, reporting that the device has a high absolute reliability (CV = 2.26–2.51%) and an extremely high relative reliability in the evaluation of peak force (ICC = 0.94–0.95). In another study, del-Cuerpo et al. [13] obtained similar results to those of this study. In their study, they evaluated two different FEMD controlled squat training protocols in a group of healthy young adults, reporting a null effect size (ES = 0.07–0.17), an acceptable absolute reliability for the protocols used (CV < 10%) for all variables, and an almost perfect relative reliability (ICC = 0.91–1.00). They also observed a low systematic bias, a coefficient of determination R^2^ = 0.988–0.518, and a level of association classified between large and almost perfect (r = 0.72–1.00).

When compared to a study that assessed the trunk rotator muscles’ reliability using FEMD in isometric mode [26], this one produced a better result. That study found that the maximal muscular force for the horizontal cable woodchop exercise was acceptable for both the dominant side (CV = 12.06%; ICC = 0.80) and the non-dominant side (CV = 17.06%; ICC = 0.60). They also assessed the low cable woodchop exercise and found that it was reliable for both the dominant side (CV = 19.67%; ICC = 0.72) and the non-dominant side (CV = 16.73%; ICC = 0.79). The reason for the discrepancy between our study and the other may be attributed to the increased stability that comes from measuring muscular strength in a seated position. This study produced very similar relative reliability results to the other study [27], which measured peak isometric strength in seated knee extension using a device and reported a high relative reliability (ICC = 0.98). This measure is linked to human locomotor efficiency and stability, and it is positively associated with quality of life [28,29].

In order to guarantee measurement reproducibility in the assessment of various force manifestations, these novel functional electromechanical devices require a sufficient familiarization procedure [30]. Studying a dependable assessment approach using FEMD may help gain a better understanding of peak force, motor gesture execution speed, and muscular power. Our study measured peak strength while the subjects were seated, which may be helpful for future research in para-Olympic athletes with limited mobility. It may also be used as a measure of peak muscular strength in studies involving sedentary or unskilled subjects or in individuals who have balance issues, such as the elderly or people with any condition that affects postural stability. Even though this study showed how reliable the FEMD is, there are several limitations that should be considered in future studies. The subjects evaluated in this study are university students, and this is a limitation that must be taken into account when comparing the findings with other populations or other methods and evaluation protocols used to evaluate isometric and isometric/vibratory strength.

## 5. Conclusions

The main findings of this study show that isometric and isometric/vibratory peak muscular strength in bilateral seated bench press, bilateral seated rowing, and unilateral seated right and left knee extension tests can be measured with high reliability and reproducibility using the FEMD to monitor them. This facilitates measurement and provides an additional and more cost-effective way to record various measures related to maximal muscle strength.

## Figures and Tables

**Figure 1 sports-12-00175-f001:**
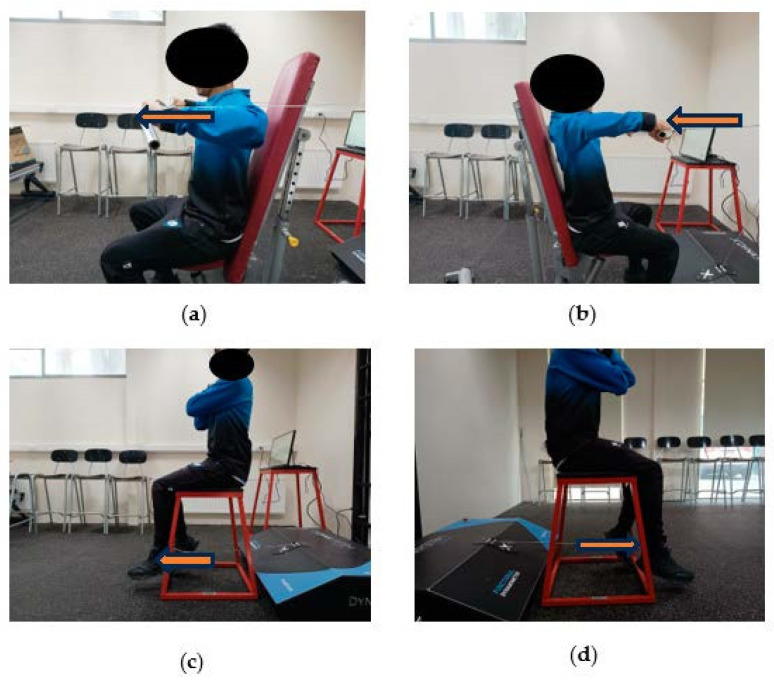
Test position adopted for (**a**) bilateral seated bench press with shoulder abduction at 90° and elbow flexion at 90°, (**b**) bilateral seated rowing with shoulder abduction at 90° and elbow flexion at 90°, (**c**) unilateral seated right knee extension in open kinetic chain with 90° knee flexion, (**d**) unilateral seated left knee extension in open kinetic chain with 90° knee flexion.

**Figure 2 sports-12-00175-f002:**
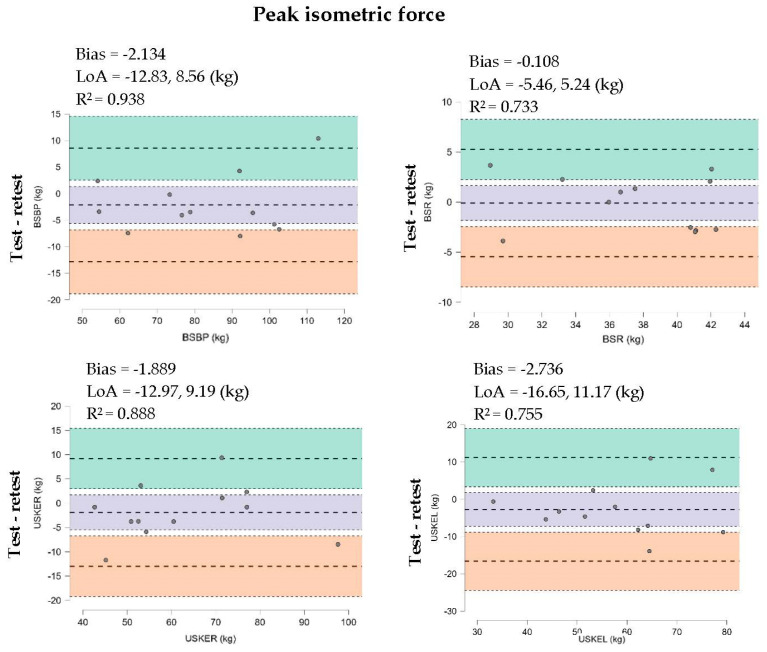
Bland-Altman test-retest plots for isometric (5 s) and iso-metric/vibratory (5 s) (Hz/mm) peak force for the bilateral seated bench press (BSBP), bilateral seated rowing (BSR), unilateral seated right knee extension (USKER), and unilateral seated left knee extension (USKEL) tests.

**Figure 3 sports-12-00175-f003:**
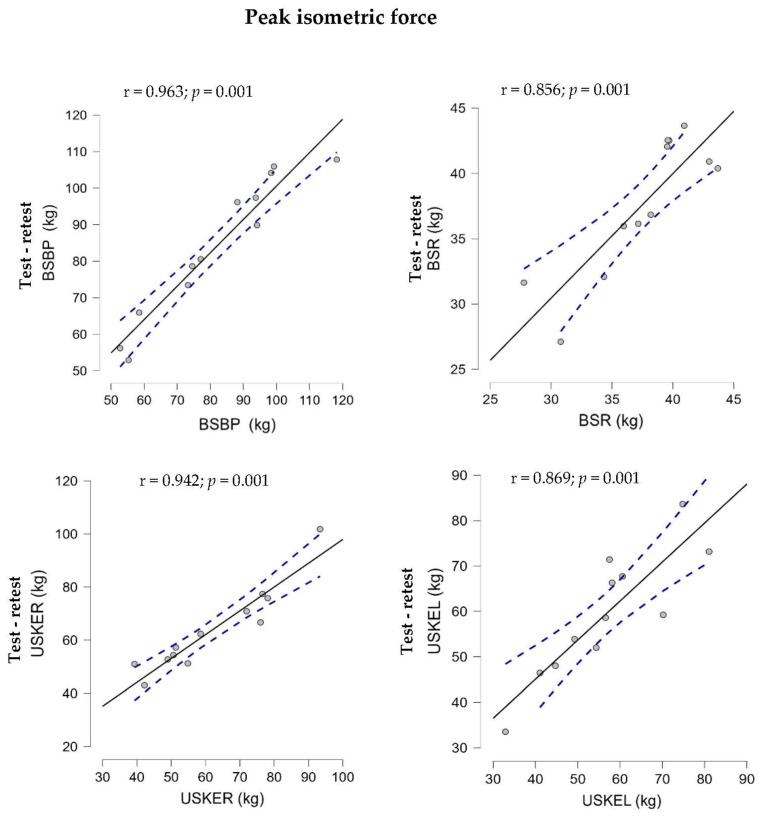
Relationship between isometric (5 s) and isometric/vibratory (Hz/mm) peak muscular strength for the bilateral seated bench press (BSBP), bilateral seated rowing (BSR), unilateral seated right knee extension (USKER), and unilateral seated left knee extension (USKEL) tests between both test sessions during maximal muscular strength assessment using an FEMD. IV: vibratory index.

**Table 1 sports-12-00175-t001:** Relative and absolute reliability for maximal isometric muscular strength.

	Mean ± SD (kg)						*p*-Value	ES	SEM	CV%	ICC
	Test			Re-Test					(95%CI)	(95%CI)	(95%CI)
BSBP	82.61	±	20.52	84.08	±	19.16	0.34	0.07	3.56 (2.73–6.55)	4.65 (3.29–7.89)	0.99 (0.95–0.99)
BSR	37.54	±	4.37	37.65	±	6.35	0.94	0.01	3.61 (2.56–6.14)	9.61 (6.81–9.81)	0.92 (0.74–0.98)
USKER	61.84	±	16.88	63.73	±	16.07	0.27	0.08	4.00 (2.83–6.79)	6.37 (4.51–10.81)	0.98 (0.92–0.99)
USKEL	56.93	±	13.93	59.49	±	13.79	0.09	0.14	5.02 (3.56–8.52)	8.63 (6.12–14.66)	0.95 (0.84–0.99)

BSBP: bilateral seated bench press; BSR: bilateral seated row; USKER: unilateral seated knee extension right; USKEL: unilateral seated knee extension left; kg: kilograms; *p*-value: significance level; SD: standard deviation; ES: Cohen’s d effect size; SEM: standard error of measurement; CV%: coefficient of variation; ICC: intraclass correlation coefficient; 95%CI: 95% confidence interval.

**Table 2 sports-12-00175-t002:** Relative and absolute reliability for maximum vibratory isometric strength 20 vibratory index (Hz/mm).

	Mean ± SD (kg)						*p*-Value	ES	SEM	CV%	ICC
	Test			Re-Test					(95%CI)	(95%CI)	(95%CI)
BSBP	87.64	±	17.99	91.11	±	22.45	0.08	0.10	4.43 (3.14–7.52)	4.95 (3.51–8.40)	0.98 (0.95–0.99)
BSR	42.77	±	8.13	42.96	±	8.05	0.89	0.01	3.59 (2.55–6.10)	8.38 (5.94–14.23)	0.95 (0.82–0.98)
USKER	65.68	±	17.06	68.38	±	18.33	0.12	0.11	4.75 (3.37–8.07)	7.09 (5.02–12.04)	0.97 (0.90–0.99)
USKEL	60.26	±	19.06	61.60	±	13.93	0.52	0.06	5.01 (3.55–8.51)	8.23 (5.83–13.97)	0.96 (0.87–0.99)

BSBP: bilateral seated bench press; BSR: bilateral seated row; USKER: unilateral seated knee extension right; USKEL: unilateral seated knee extension left; kg: kilograms; *p*-value: significance level; SD: standard deviation; ES: Cohen’s d effect size; SEM: standard error of measurement; CV%: coefficient of variation; ICC: intraclass correlation coefficient; 95%CI: 95% confidence interval.

**Table 3 sports-12-00175-t003:** Relative and absolute reliability for maximum vibratory strength 40 vibratory index (Hz/mm).

	Mean ± SD (kg)						*p*-Value	ES	SEM	CV%	ICC
	Test			Re-Test					(95%CI)	(95%CI)	(95%CI)
BSBP	97.19	±	21.72	100.24	±	23.40	0.15	0.09	4.96 (3.51–8.41)	5.02 (3.56–8.52)	0.98 (0.95–0.99)
BSR	48.03	±	8.91	48.43	±	8.13	0.77	0.03	3.46 (2.45–5.87)	7.17 (5.08–12.18)	0.96 (0.87–0.99)
USKER	71.92	±	16.13	72.01	±	14.94	0.95	0.01	4.93 (3.49–8.37)	4.93 (3.49–8.37)	0.98 (0.94–0.99)
USKEL	69.00	±	16.48	69.49	±	14.20	0.75	0.02	5.35 (3.79–9.09)	3.71 (2.63–6.29)	0.98 (0.94–0.99)

BSBP: bilateral seated bench press; BSR: bilateral seated row; USKER: unilateral seated knee extension right; USKEL: unilateral seated knee extension left; kg: kilograms; *p*-value: significance level; SD: standard deviation; ES: Cohen’s d effect size; SEM: standard error of measurement; CV%: coefficient of variation; ICC: intraclass correlation coefficient; 95%CI: 95% confidence interval.

## Data Availability

All relevant data are within the manuscript. The datasets generated during and/or analyzed during the current study are available from the corresponding author upon reasonable request.

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
