# Peer review of "Test–Retest Reliability of an Isometric and Isometric/Vibratory Muscular Strength Protocol with Functional Electro-Mechanical Dynamometry"

_sports, 2024, doi:10.3390/sports12070175_

Round 1
Reviewer 1 Report
Comments and Suggestions for Authors
The use of so many abbreviations in the abstract makes it difficult to read and understand.
Line 77: DEMF? Or FEMD?
The sample of 12 subjects seems insufficient to me.
It would be good to explain the tool (FEMD) used in more detail. Many readers do not know it
The figures should be better explained
You should emphasize more the importance of this article for the scientific community. What does it really contribute?
Comments on the Quality of English LanguageNo comments
Author Response
Dear Corrector,
I would like to thank you for your comments and send you the answers to your requests in the attached file.
Thank you very much.

Reviewer 2 Report
Comments and Suggestions for Authors
The scientific substantiation of algorithms for controlling various aspects of human physical fitness has been an important area of scientific research for a long time.
This study scientifically proves the possibility of conducting a set of tests to assess the strength abilities of human muscles. The use of modern research methods and thorough statistical analysis is impressive.
The article describes in an accessible and gradual manner the sequence of solving the research objective. The study is focused on a detailed consideration of relevant, up-to-date research (mainly over the past 5 years).
However, in the course of reviewing the article, some issues arose that require additional consideration:
- It is undesirable to use abbreviations in the abstract;
- in the introduction, the authors should present the results of other researchers in the chosen area of research more widely, which will allow them to clearly form a list of issues that still need to be addressed;
- The authors should explain the reasons for the choice of tests for the study;
- line 94 - 95, it is necessary to indicate the units of measurement of indicators, and pay attention to the value of body length ("height 1.71 ± 2.82");
- lines 105 - 113, 142 - 165 should be updated to reflect the similarity of the information presentation with another scientific paper(https://www.mdpi.com/1660-4601/20/2/1289);
- the calculation of coefficient of variation (CV) and the values it acquires in the course of the study should be clarified (e.g., Table 1, USKEL - 14.66);
- information should be provided in the section (2.6 Statistical analysis) on what results were obtained during the sample testing using the Shapiro-Wilk test;
- Figure 1 does not give an idea of the test exercises used in the study;
- in the notes to the tables, specify the units of measurement for all indicators;
- in the figures, add labels to the axes of the diagrams;
- in the conclusions, the authors note the possibility of using these tests for athletes, although no research was conducted in this area.
Author Response

(The authors gave the same response as above.)

Reviewer 3 Report
Comments and Suggestions for Authors
Rev sports-2914761
Test-retest reliability of an isometric and isometric/vibratory muscular strength protocol with functional electro-mechanical dynamometry
Dear Authors
Thank you for the opportunity to review the article sports-2914761 - Test-retest reliability of an isometric and isometric/vibratory muscular strength protocol with functional electro-mechanical dynamometry, carried out to analyze the test-retest reliability of an isometric and isometric/vibratory muscular strength protocol in the BSBP, BSR, USKER, and USKEL tests controlled by FEMD in healthy young adults.
The work presented falls within the journal's scope, and it will be helpful to readers.
This methodological work is generally well framed and designed, with the method, analysis of results, and discussion of the reliability of physical tests being common in methodological articles.
Given the above, I only have small comments/suggestions to clean up some errors/inaccuracies, mainly in the writing.
Kind regards
_______________
Specific comments
Introduction
L41-45. “… strength [1–7]. Which allows better analysis and precision in the evaluation of training systems and changes in the level of physical performance that induce muscular strength training, which can favor and optimize the function of coaches and trainers. when providing feedback to the training process and competition [8].” // Suggestion: “… strength [1–7], which allows (i) better analysis and precision in the evaluation of training systems and changes in the level of physical performance that induce muscular strength training, and (ii) can favor and optimize the function of coaches and trainers, when providing feedback to the training process and competition [8].”
L65. “with coefficient of variation (CV) < 11.97 percent.” // “with CV < 11.97%.”
L65-66. “According to the study Baena-Raya et al. [16]” // “According to Baena-Raya et al. [16]”
L67-68. “In the study Jerez-Mayorga et al. [14]” // “In the Jerez-Mayorga et al. [14] study”
L94. “12 physically” // “Twelve (n = 12) physically”
L94-95. Present units for weight and height.
L99-100. “The Universidad Católica de la Santísima Concepción No. 47/2023 ethics and research committee authorized the study protocol” // Present the authorization date.
L169: “p” // “p” – extend to all the document, i.e. p-value in italics.
L302. “R2” // “R2”
L311. “(CV = 3.90-11.57 and …” // “(CV = 3.90-11.57%) and …”
L312. “In the study Rodriguez-Perea et al. [24] showed…” // “Rodriguez-Perea et al. [24] showed…”
L316. “(ICC = 0. 74-0.87), in addition, …” // “(ICC = 0. 74-0.87). In addition, …”
L318. “(CV = 7.04-14.00%), in addition, in this…” // “(CV = 7.04-14.00%). In this…” (start a new paragraph with: In this study…
L320. “In another study Martinez-Garcia et al [25]”// “In another study, Martinez-Garcia et al. [25]”
L323. “CV = 7.28 %” // “CV = 7.28%” Also in L324, L326, L328.
L344. “r” // “r”. (italic)
L348-349. “(CV = 17.06 percent; …” // “(CV = 17.06%; …”
Comments on the Quality of English LanguageNo issues detected.
Author Response
Dear proofreader:
Along with greetings and thanks for your corrections, I am sending the requested modifications in an attached file.
Regards

Round 2
Reviewer 2 Report
Comments and Suggestions for Authors
3- In the introduction, the authors should present the results of other researchers in the chosen area of research more widely, which will allow them to clearly form a list of issues that still need to be addressed.
please indicate where the changes were made
4- Line 94 - 95, it is necessary to indicate the units of measurement of indicators, and pay attention to the value of body length ("height 1.71 ± 2.82")
Line 98 Please specify ± 2.82 m? Body length 2 metres 82 centimetres.
Author Response
Dear proofreader
Thank you very much for your indications, please find attached a document with the requested changes.
Regards
